# Comparing Diagnostic Performance of Short and Long [^18^F]FDG-PET Acquisition Times in Giant Cell Arteritis

**DOI:** 10.3390/diagnostics14010062

**Published:** 2023-12-27

**Authors:** Pieter H. Nienhuis, Marieke van Nieuwland, Gijs D. van Praagh, Karolina Markusiewicz, Edgar M. Colin, Kornelis S. M. van der Geest, Nils Wagenaar, Elisabeth Brouwer, Celina Alves, Riemer H. J. A. Slart

**Affiliations:** 1University of Groningen, University Medical Center Groningen, Medical Imaging Center, Department of Nuclear Medicine and Molecular Imaging, 9713 GZ Groningen, The Netherlands; 2Hospital Group Twente, Department of Rheumatology and Clinical Immunology, 7600 SZ Almelo, The Netherlands; m.vnieuwland@zgt.nl (M.v.N.); c.alves@zgt.nl (C.A.); 3University of Groningen, University Medical Center Groningen, Department of Rheumatology and Clinical Immunology, 9713 GZ Groningen, The Netherlands; 4Medical University of Warsaw, 02-091 Warsaw, Poland; 5Hospital Group Twente, Department of Nuclear Medicine, 7555 DL Hengelo, The Netherlands; 6University of Twente, Faculty of Science and Technology, Department of Biomedical Photonic Imaging, 7522 NB Enschede, The Netherlands

**Keywords:** large-vessel vasculitis, giant cell arteritis, positron emission tomography, fluorodeoxyglucose, vascular inflammation, diagnostic accuracy, inter-observer agreement, intra-observer agreement

## Abstract

(1) Background: In giant cell arteritis (GCA), the assessment of cranial arteries using [^18^F]fluorodeoxyglucose ([^18^F]FDG) positron emission tomography (PET) combined with low-dose computed tomography (CT) may be challenging due to low image quality. This study aimed to investigate the effect of prolonged acquisition time on the diagnostic performance of [^18^F]FDG PET/CT in GCA. (2) Methods: Patients with suspected GCA underwent [^18^F]FDG-PET imaging with a short acquisition time (SAT) and long acquisition time (LAT). Two nuclear medicine physicians (NMPs) reported the presence or absence of GCA according to the overall image impression (gestalt) and total vascular score (TVS) of the cranial arteries. Inter-observer agreement and intra-observer agreement were assessed. (3) Results: In total, 38 patients were included, of whom 20 were diagnosed with GCA and 18 were without it. Sensitivity and specificity for GCA on SAT scans were 80% and 72%, respectively, for the first NMP, and 55% and 89% for the second NMP. On the LAT scans, these values were 65% and 83%, and 75% and 83%, respectively. When using the TVS, LAT scans showed especially increased specificity (94% for both NMPs). Observer agreement was higher on the LAT scans compared with that on the SAT scan. (4) Conclusions: LAT combined with the use of the TVS may decrease the number of false-positive assessments of [^18^F]FDG PET/CT. Additionally, LAT and TVS may increase both inter and intra-observer agreement.

## 1. Introduction

Giant cell arteritis (GCA) is a vasculitis of medium- to large-sized arteries that can lead to severe complications, when treatment is delayed, such as irreversible blindness or aneurysms [1,2]. Therefore, the early and accurate diagnosis and treatment of GCA patients is important [3]. Nevertheless, delays in GCA diagnosis are not uncommon [4,5]. GCA occurs mainly in patients aged 50 years and older with a peak incidence for those aged 72 years. GCA is a medical emergency as severe complications include stroke, permanent vision loss and aneurysms that can arise within days. Early treatment with high-dose glucocorticoids (GCs) is needed for GCA management, and prognosis depends on rapid GC treatment and the severity of symptoms at disease onset [6]. Typical symptoms include new headache, jaw claudication and/or scalp tenderness, but more general constitutional symptoms such as fatigue, fever and weight loss are also common. As GCA patients often present with nonspecific signs and symptoms, this leads to challenges in early recognition and diagnosis [3]. Also, unnecessary GC treatment should be avoided as long-term adverse effects and toxicity are often seen [7]. Recommendations by the European Alliance of Associations for Rheumatology (EULAR) state that additional testing via imaging, preferably with ultrasound as a first test, or histology is needed to confirm or rule out GCA after clinical presentation [8].

GCA is a heterogeneous disease that can manifest exclusively in cranial arteries (C-GCA), extracranial large vessels (LV-GCA) or both (C/LV-GCA) [9]. Arteries commonly involved in C-GCA are temporal, maxillary, facial, and occipital arteries. The thoracic aorta, subclavian, and carotid arteries are commonly affected in LV-GCA [10]. Traditionally, a temporal artery biopsy (TAB) is performed for GCA diagnosis; however, this method comes with limitations such as low sensitivity (reported to be 39%) [11]. Therefore, imaging modalities are increasingly used in the diagnosis of GCA [8,12]. Whole-body [^18^F]fluorodeoxyglucose ([^18^F]FDG) positron emission tomography (PET) combined with low-dose computed tomography (CT) is one of the imaging methods used in the diagnosis of GCA. Active inflammatory cells have increased glucose metabolism, which can be visualized as increased [^18^F]FDG uptake in the inflamed vessel wall [12,13,14]. 

The use of [^18^F]FDG PET/CT is mostly described as suitable for LV-GCA as opposed to C-GCA because cranial arteries are relatively small [15]. However, technical improvements such as the digital PET camera system and innovative image reconstruction of conventional PET camera data have resulted in increased spatial resolution in PET imaging. Hence, the imaging of the cranial arteries in C-GCA has become feasible [15]. Using [^18^F]FDG PET/CT as an imaging modality for C-GCA diagnosis is of clinical value as ultrasonography as a first diagnostic tool is not always conclusive and depends on availability [11]. Recently, several studies have reported the good diagnostic performance of [^18^F]FDG PET/CT for C-GCA, with a sensitivity of 79% and a specificity of 92% [15]. This makes [^18^F]FDG PET/CT suitable to detect both C-GCA and LV-GCA, which increases its value in early GCA diagnosis [16]. 

Besides better PET camera systems and better image reconstruction, the detectability of inflammation in the cranial arteries may be improved via prolonging image acquisition times due to higher-quality imaging, as recommended by EULAR [8,17]. However, whether or not this improves diagnostic performance has not yet been investigated. Furthermore, if this is true, it would be beneficial for use in numerous hospitals using older PET camera systems, which do not have access to the newest (digital) state-of-the-art PET systems. A comparison of diagnostic performance between a prolonged 5 min acquisition time (long acquisition time (LAT)) and a 2 min regular acquisition time is lacking (short acquisition time (SAT)). In this study, we aim to investigate the diagnostic performance of a prolonged [^18^F]FDG-PET/CT acquisition time in terms of the detectability of GCA compared that under the standard acquisition time in patients with suspected GCA.

## 2. Materials and Methods

### 2.1. Study Design

Patients with suspected GCA were recruited from a Hospital Group TwenteGCA Early in Twente (ZGT GET) cohort upon the first visit between July 2020 and September 2021. In a nested case–control study, we intended to include 20 GCA patients and 20 patients without GCA consecutively based on baseline diagnosis (submitted data). The ZGT GET cohort prospectively follows patients for five years to study disease progression in a fast-track clinic. Treatment for GCA was not started before diagnostic work-up, but was not delayed when deemed necessary by the treating rheumatologist before imaging. For the original nested case–control study, inclusion criteria were an age ≥ 50 years and ability to understand study information. An exclusion criterion was the presence of contra-indications for imaging such as claustrophobia. Patients in the nested case–control study who did not undergo LAT scanning of the head and neck area were excluded for analyses. Patients were followed for six months to verify the diagnosis of GCA. The study was approved by the medical ethical committee (MEC-U) and was performed in accordance with the Declaration of Helsinki. Written consent was obtained from all patients. This study was a sensitivity analysis focusing on [^18^F]FDG PET/CT in the patient population of this nested case–control study. 

### 2.2. Imaging and Assessment

Whole-body [^18^F]FDG PET/CT (Siemens Biograph 40 mCT Flow, Siemens Healthineers, Knoxville, TN, USA) was performed within five working days after the diagnostic work-up. All patients were injected with 2–3 MBq/kg [^18^F]FDG, and 60 min post-injection, they underwent [^18^F]FDG PET/CT whole-body imaging of the head with an LAT, and with a 0.3 mm/sec table motion, and of the head, neck and whole body region, with an SAT, and with 1 mm/s continuous motion of the patient table. When comparing this to a ‘step-and-shoot’ scan method, the LAT of the head was approximately 12.8 min. The SAT was approximately 4.2 min for the head from a total of approximately 16.6 min for whole-body acquisition. 

For the purpose of this study, imaging was retrospectively assessed by two independent nuclear medicine physicians (NMPs; RS (~15 years of experience of PET imaging in GCA) and NW (~15 years of experience of all-round PET imaging)). The NMPs were blinded to any patient information, including final diagnosis, to avoid bias. Additionally, a washout period of five months was used between the LAT and SAT scan assessment. To look at intra-observer agreement, scans of seven patients were randomly selected to be assessed twice by the NMPs. LAT and SAT scans were independently assigned as positive or negative for GCA by the NMPs based on the gestalt (overall impression) of the entire scans, including whole-body large vessels [13]. Additionally, [^18^F]FDG uptake was quantitatively assessed in LAT-accessible arteries in the head and neck area, these being the superficial temporal (TA), maxillary (MA), and vertebral arteries (VA), and scored using previously described methods [15,18]. A score between 0 and 2 was used with 0 defined as no [^18^F]FDG uptake higher than that in the surrounding tissue, 1 defined as [^18^F]FDG uptake slightly higher than that in the surrounding tissue, and 2 defined as [^18^F]FDG uptake significantly higher than that in the surrounding tissue. For each patient, a total vascular score (TVS) was calculated by summing up the scores of the cranial arteries. As such, the TVS ranged from 0 to 12.

Lastly, [^18^F]FDG uptake was assessed semi-quantitively by drawing regions of interest (ROI) around areas of increased uptake in the cranial arteries (TA, MA, and VA). The highest [^18^F]FDG uptake measured in a maximum standardized uptake value corrected for lean body mass (SULmax) was recorded for each artery on the SAT and LAT scans. Semi-quantitative analysis also included 7 scans that were included twice, as described above [19,20].

### 2.3. Case Definition

The gold standard in GCA diagnosis is debatable. Therefore, two experts in the GCA field (EC and KvdG) assessed patient information to conclude a clinical diagnosis based on clinical parameters after six months of follow-up. Experts received information on signs and symptoms, physical examination, laboratory values (C-reactive protein and erythrocyte sedimentation rate), medication use, and the presence of an alternative diagnosis at the baseline visit and the 6-month visit. They were blinded for all imaging and temporal artery biopsy results and were not involved in diagnostic work-up or patient care. In the case of discrepancies, a third expert (EB) was involved to make a definitive decision. This clinical diagnosis made by the expert panel after six months of follow-up, blinded for imaging results, was used as a reference standard in this study. 

### 2.4. Statistical Analysis

Mean values with standard deviation (SD) or median values with interquartile ranges (IQR) were used to describe baseline characteristics when appropriate after testing for normality. To assess diagnostic performance, sensitivity and specificity were calculated via cross-tabulation for the gestalt assessment. Secondly, a receiver operating characteristic (ROC) analysis was performed to determine the optimal TVS and SULmax using the Youden index. Based on this, cross-tabulation was used to calculate the sensitivity and specificity of the TVS. Inter- and intra-observer variability were calculated using Fleiss Kappa as described before, with similar terminology [21]. Data are represented as means and 95% confidence intervals for normally distributed data, and as medians and interquartile ranges for non-normally distributed data.

## 3. Results

### 3.1. Patient Characteristics

From the original nested case–control study (*n* = 42), four patients were excluded from analysis as LAT was not performed. In total, 38 patients were included in this study, of whom 20 were diagnosed with GCA and 18 were suspected of having but not diagnosed with GCA based on the reference standard (clinical diagnosis by the expert panel after six months of follow-up). The mean age was 72.1 ± 7.0 years in the GCA group and 70.1 ± 8.8 years in the no GCA group. Of the 20 GCA patients, 18 met the 2022 ACR/EULAR GCA classification criteria [22]. Patient characteristics are described in Table 1.

### 3.2. Diagnostic Performance of SAT vs. LAT

The diagnostic performance of the gestalt assessment of the two NMPs is shown in Table 2a. NMPs showed divergent sensitivity and specificity for the SAT scans. Sensitivity and specificity were, respectively, 80% and 72% for NMP #1, and 55% and 89% for NMP #2. When assessing the LAT scans, sensitivities were 65% and 75%, and specificities were similar between NMPs (83%).

Figure 1 gives an overview of the 0–2 visual uptake scores, and Figure 2 demonstrates an example of SAT and LAT images of one patient. Receiver operating characteristic (ROC) analysis revealed areas under the curve for the TVS for the two NMPs of 0.84 (0.73–0.95, *p* < 0.001) and 0.82 (0.70–0.94, *p* < 0.001) for the SAT scans, and of 0.85 (0.73–0.96, *p* < 0.001) and 0.77 (0.64–0.90, *p* = 0.001) for the LAT scans. The optimal TVS cut-off based on the Youden index was a score of 1. Table 2b shows the diagnostic performance when using this ROC-based TVS cut-off score. For the SAT scans, sensitivity and specificity were, respectively, 70% and 83% for NMP #1, and 65% and 89% for NMP#2. For the LAT scans, sensitivities were 70% and 60% for the NMPs, respectively, and specificity was 94% for both NMPs.

The SULmax of the MA, TA and VA are shown in Figure 3 for an SAT and LAT. The median SULmax was increased under an LAT compared to SAT for patients with GCA and those without. For GCA patients, the median SULmax in the MA increased from 1.4 (IQR 1.0–2.1) to 1.5 (IQR 1.1–2.7) (*p* < 0.0001), that in the TA increased from 1.2 (IQR 1.0–1.4) to 1.4 (IQR 1.1–1.8) (*p* < 0.0001), and that in the VA increased from 1.6 (IQR 1.3–2.1) to 2.0 (IQR 1.4–2.8) (*p* < 0.0001). For patients without GCA, the median SULmax in the MA increased from 1.0 (IQR 0.8–1.3) to 1.1 (IQR 0.9–1.4) (*p* < 0.0001), that in the TA increased from 1.0 (IQR 0.8–1.2) to 1.1 (IQR 0.8–1.3) (*p* < 0.0001), and that in the VA increased from 1.2 (IQR 1.0–1.5) to 1.3 (IQR 1.1–1.5) (*p* < 0.0001). Figure 4 shows boxplots of the highest SULmax values per patient (max SULmax) separately for SAT and LAT, and for patients with and without GCA. A Mann–Whitney U test showed significant differences between patients with GCA (GCA+) and without (GCA−) for both an SAT (*p* = 0.0002) and LAT (*p* < 0.0001). For GCA patients, no significant differences were found between an SAT and LAT whereas this was found for GCA+ patients (*p* = 0.0475).

The diagnostic performance of the SULmax measurements were analyzed using an ROC analysis. The area under the curve in the arteries for an SAT and LAT, respectively, were 0.77 (0.68–0.86, *p* < 0.0001) and 0.79 (0.70–0.88, *p* < 0.0001) for the MA, 0.63 (0.52–0.75, *p* = 0.0033) and 0.68 (0.57–0.79, *p* = 0.0033) for the TA, and 0.80 (0.71–0.89, *p* < 0.0001) and 0.84 (0.76–0.92, *p* < 0.0001) for the VA. The highest SULmax in each patient (max SULmax) had an area under the curve of 0.81 (0.69–0.93, *p* = 0.0004) for the SAT scan and 0.86 (0.76–0.97, *p* < 0.0001) for the LAT scan. Using a Youden index-based cut-off score on this ROC curve resulted in SULmax cut-offs of 1.35 on the SAT scan and 1.95 on the LAT scan. The sensitivity and specificity obtained using these scores is shown in Table 3.

### 3.3. Inter-Observer Agreement

A moderate agreement was found between both NMPs’ gestalt assessments on the SAT scans (Fleiss Kappa: 0.49 [0.24–0.74]). The agreement was substantial for the LAT scans (Fleiss Kappa: 0.79 [0.59–0.98]). When using a TVS cut-off of 1, the agreement was substantial (0.79 [0.59–0.98]) for the SAT scans and almost perfect (0.89 [0.74–1.00]) for the LAT scans.

### 3.4. Intra-Observer Agreement

For gestalt scoring, the intra-observer agreement for the SAT scans was substantial (0.72 [0.23–1.00]) for NMP #1 and fair (0.30 [0.47–1.00]) for NMP #2. Intra-observer agreement increased for the LAT scans to full agreement (1.00 [0.72–1.00]) for NMP #1 and to substantial (0.72 [0.23–1.00]) for NMP #2. 

When considering a TVS cut-off of 1 for the cranial arteries as positive, the intra-observer agreement for NMP #1 on the SAT scan was also substantial (0.72 [0.23–1.00]). Moreover, NMP #1 had full agreement when using the TVS for the LAT scan. Full agreement (1.00) was found for the TVS on both SAT and LAT scans assessed by NMP #2.

For the semi-quantitative assessment, intra-observer agreement was assessed using the previously ROC-defined cut-offs for SULmax values. Any cranial artery with a SULmax value exceeding 1.35 (SAT) or 1.95 (LAT) was considered positive. This resulted in a moderate intra-observer agreement of (0.46 [−0.07–0.99] on the SAT scans. On the LAT scans, full intra-observer agreement was obtained (1.00). 

## 4. Discussion

In this study, we compared the diagnostic performance under a prolonged [^18^F]FDG-PET/CT acquisition time to that under a regular acquisition time in patients with suspected GCA. It was demonstrated that the inter- and intra-observer agreement of NMPs increased using a prolonged [^18^F]FDG-PET/CT acquisition time of five minutes, or an LAT, compared with that using an SAT. Furthermore, in our study, the specificity of [^18^F]FDG-PET/CT improved when the NPs used a visual uptake scoring method (TVS) or semi-quantitative assessment compared with that when they performed gestalt assessment. These findings are of clinical importance to better apply [^18^F]FDG-PET/CT for the diagnosis of cranial GCA. 

LAT scanning is recommended by the EANM and the EULAR [8,17]. Our study showed that this results in a better inter- and intra-observer agreement and therefore a more reliable assessment of [^18^F]FDG-PET/CT considering the head and neck area compared to that under an SAT. Using an LAT with the TVS or semi-quantitative assessment, the sensitivity and specificity of both readers were comparable to those in the previous literature [15,23,24,25]. As known, [^18^F]FDG-PET/CT has an important role in diagnosing extracranial vascular involvement [17,26,27]. In this study, the maxillary, temporal and vertebral arteries were assessed visually using both an SAT and LAT. Using the TVS, diagnostic performance improved compared with that under gestalt assessment even though the latter includes an assessment of the extracranial vessels as well. Although it was demonstrated that first gestalt assessment can have good diagnostic value and there is no strong evidence that the semi-quantitative or TVS should replace gestalt assessment in practice, our data indicate that using a standardized visual score to assess arteries in the entire body, including extracranial arteries such as the aorta and its branches, might increase diagnostic performance even further [13,25]. The standardization of image assessment in [^18^F]FDG-PET/CT also led to better observer agreement in our study, supporting the previous literature [28]. Furthermore, semi-quantitative assessment resulted in similar diagnostic performance to that when using a TVS, although sensitivity may have been slightly lower on the LAT scans. Importantly, using a semi-quantitative assessment resulted in markedly increased specificity when using the LAT scans compared to that when using the SAT scans. Full intra-observer agreement was also found when using the LAT scans whereas it was only moderate on the SAT scans, further strengthening the results for the TVS values.

Considering the location and size of the vertebral artery compared to those of the temporal and maxillary artery, classifying patients with vertebral artery involvement as C-GCA or LV-GCA can be debated. Nevertheless, [^18^F]FDG uptake in the vertebral artery can be assessed using the prolonged acquisition time of the head and neck area. Hence, vertebral artery evaluation contributed to the comparison between SAT and LAT (see Figure 2). Interestingly, the involvement of the vertebral artery in GCA is described to be rare [29,30], which is not supported in this study as around half of our patients had vertebral artery involvement upon [^18^F]FDG-PET/CT. This observation was shown in previous research in a different cohort of our group as well [15]. 

This study provides a unique dataset of GCA patients and patients suspected of have GCA that were not diagnosed with GCA where both SAT and LAT scans were performed for every patient, which has never been carried out before to our knowledge. [^18^F]FDG PET/CT was performed by experienced professionals and retrospectively assessed by expert NMPs in the field of GCA. Additionally, the use of an [^18^F]FDG PET/CT scanner in a peripheral hospital closely relates to clinical practice, as state-of-the-art technology that is mainly used in studies is often not available in the majority of clinical settings. The main limitation in this study is its small study population. Nevertheless, this study provides unique results for a rare disease. Furthermore, glucocorticoid treatment was already initiated after standard diagnostic work-up when deemed necessary by the treating rheumatologist. This again makes our results applicable to clinical practice as medication should be started with short notice to prevent severe complications, and the direct availability of [^18^F]FDG PET/CT is not guaranteed. Also, in our data, glucocorticoid use did not affect diagnostic performance. Last, diagnostic performance might have been affected by the blinding of the NMPs, as knowledge of clinical characteristics could possibly increase diagnostic performance [31]. 

## 5. Conclusions

In conclusion, the assessment of the head and neck region for cranial GCA can be improved by using a prolonged [^18^F]FDG-PET/CT acquisition time. However, based on the gestalt assessment, this does not directly reflect better diagnostic performance, as sensitivity and specificity did not improve when using an LAT in comparison to those when using an SAT. Further research could focus on comparing gestalt assessment with the visual uptake scoring of both cranial and extracranial arteries in a larger cohort, as our results indicate slight improvement in diagnostic performance. Nevertheless, an LAT improved intra- and inter-observer agreement and does not require excessive additional time and costs; hence, it can be of value in general imaging practices.

## Figures and Tables

**Figure 1 diagnostics-14-00062-f001:**
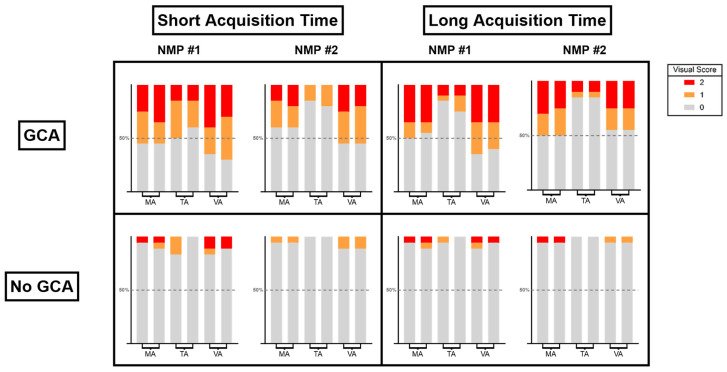
Graphical presentation of the uptake scores for the short-acquisition-time and long-acquisition-time scans for both observers. The scored cranial arteries are shown separately with the left bar and right bar representing the left and right artery, respectively.

**Figure 2 diagnostics-14-00062-f002:**
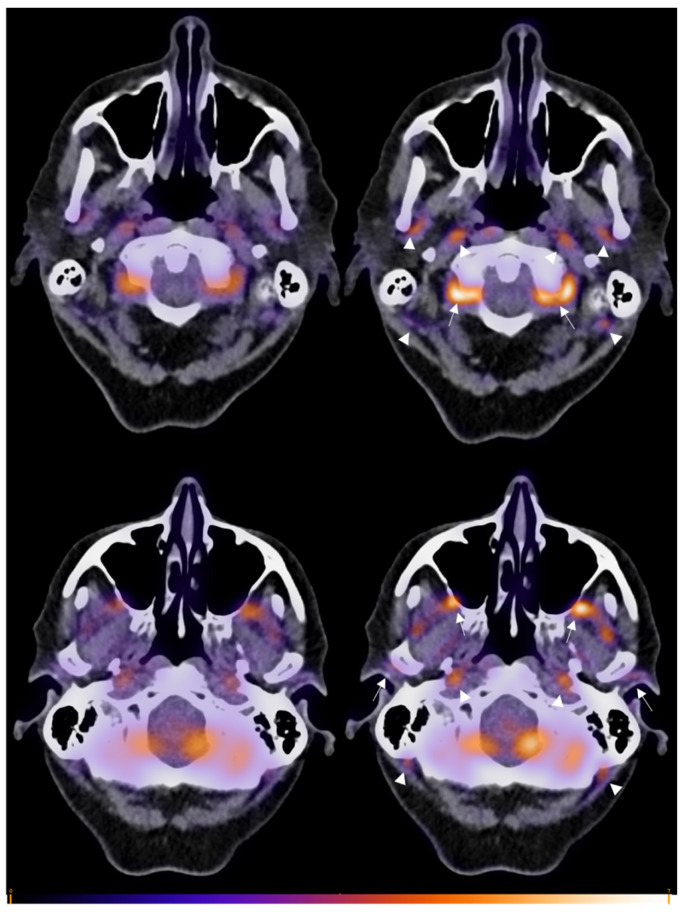
Example images of cranial artery uptake in a GCA patient within the short acquisition time of 2 min (**left**) and long acquisition time of 5 min (**right**). Uptake is visibly more intense in the long-acquisition-time scan. An uptake markedly higher than that in the background (visual score 2) is seen in the vertebral arteries (upper image, arrows) and maxillary arteries (lower image, arrows), and an uptake slightly higher than that in the background (visual score 1) can be seen in the temporal arteries (lower image, arrows). Additionally, uptake may be appreciated in the external carotid arteries (upper image, arrowheads), occipital arteries and internal carotid arteries (upper and lower images, arrowheads) but was not taken into account for the visual scoring method in this study.

**Figure 3 diagnostics-14-00062-f003:**
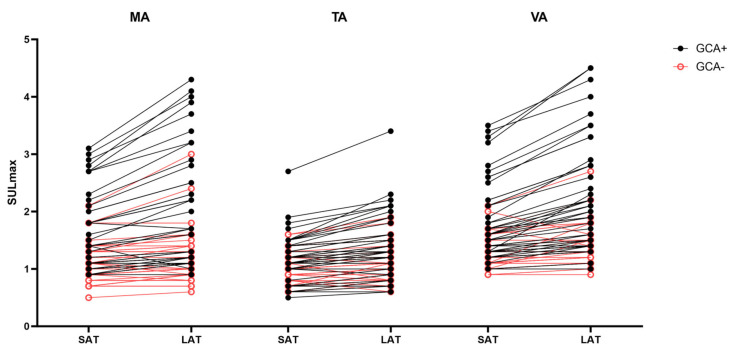
Individual SULmax values of the maxillary arteries (MA), temporal arteries (TA) and vertebral arteries (VA) for patients with GCA (GCA+) and those without GCA (GCA−). Paired scores for the short acquisition time (SAT) and long acquisition time (LAT) are connected with a line.

**Figure 4 diagnostics-14-00062-f004:**
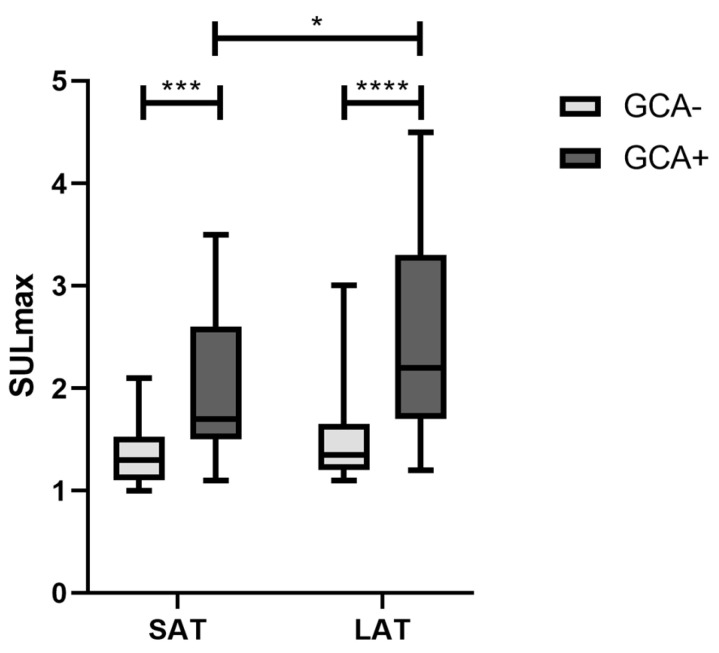
The highest SULmax values per patient (max SULmax) on short-acquisition-time (SAT) and long-acquisition-time (LAT) scans in boxplots separately for patients with GCA (GCA+) and without GCA (GCA−) according to expert panel diagnosis 6 months after diagnosis. * *p* < 0.05, *** *p* < 0.001, and **** *p* < 0.0001 (Mann–Whitney U test).

**Table 1 diagnostics-14-00062-t001:** Patient characteristics using expert panel diagnosis after six months as a reference standard.

	GCA (*n* = 20)	No GCA (*n* = 18)	*p*-Value
Age (years)	72.1 ± 7.0	70.1 ± 8.8	0.453
Females (%)	11 (55.0)	13 (72.2)	0.272
CRP (mg/L)	34.0 [10.0–44.0]	10.5 [1.75–27.8]	0.020
ESR (mm/h)	47.5 [30.3–90.3]	28.0 [13.3–49.3]	0.020
Cranial symptoms	20 (100.0)	17 (94.4)	0.474
Constitutional symptoms	16 (80.0)	12 (66.7)	0.351
Glucocorticoid use Number of days	10 (50.0) 3.5 [1.8–5.8]	0	0.018

Numbers shown as mean ± standard deviation, *N* (%), or median [interquartile range]. CRP = C-reactive protein; ESR = erythrocyte sedimentation rate.Cranial symptoms: headache, jaw claudication, visual loss and/or scalp tenderness. Constitutional symptoms: fatigue, weight loss and/or fever.

**Table 2 diagnostics-14-00062-t002:** (**a**) Cross-tabulation of the gestalt assessment for the two NMPs and the expert panel diagnosis. (**b**) Cross-tabulation of the cranial artery TVS for the two NMPs and the expert panel diagnosis. A scan was considered positive if the summed up TVS was higher than 1.

(**a**)												
	**SAT**						**LAT**					
	**NMP #1**			**NMP #2**			**NMP #1**			**NMP #2**		
	**Pos**	**Neg**		**Pos**	**Neg**		**Pos**	**Neg**		**Pos**	**Neg**	
**GCA** **(*n* = 20)**	16	4	Se 80% (58–92)	11	9	Se 55% (34–74)	13	7	Se 65% (43–82)	15	5	Se 75% (53–89)
**No GCA** **(*n* = 18)**	5	13	Sp 72% (49–88)	2	16	Sp 89% (67–98)	3	15	Sp 83% (61–94)	3	15	Sp 83% (61–94)
(**b**)												
	**SAT**						**LAT**					
	**NMP #1**			**NMP #2**			**NMP #1**			**NMP #2**		
	**Pos**	**Neg**		**Pos**	**Neg**		**Pos**	**Neg**		**Pos**	**Neg**	
**GCA** **(*n* = 20)**	14	6	Se 70% (48–85)	13	7	Se 65% (43–82)	14	7	Se 70% (48–85)	12	8	Se 60% (39–78)
**No GCA** **(*n* = 18)**	3	15	Sp 83% (61–94)	2	16	Sp 89% (67–98)	1	17	Sp 94% (74–100)	1	17	Sp 94% (74–100)

Brackets contain the 95% confidence interval of sensitivity and specificity. GCA = giant cell arteritis; SAT = short acquisition time; LAT = long acquisition time; NMP = nuclear medicine physician; Pos = positive; Neg = negative; Se = sensitivity; Sp = specificity; TVS = total vascular score.

**Table 3 diagnostics-14-00062-t003:** Cross-tabulation of the highest cranial artery SULmax and the expert panel diagnosis. A scan was considered positive if any of the arteries had a SULmax exceeding 1.35 on the SAT scan and exceeding 1.95 on the LAT scan.

	SAT			LAT		
	Pos	Neg		Pos	Neg	
**GCA** **(*n* = 20)**	18	2	Se 90% (70–98)	12	8	Se 60% (39–78)
**No GCA** **(*n* = 18)**	7	11	Sp 61% (39–80)	1	17	Sp 94% (74–100)

Brackets contain the 95% confidence intervals of sensitivity and specificity. GCA = giant cell arteritis; SAT = short acquisition time; LAT = long acquisition time; Pos = positive; Neg = negative; Se = sensitivity; Sp = specificity.

## Data Availability

The data presented in this study are available on request from the corresponding author.

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
