# Peer review of "Comparing Diagnostic Performance of Short and Long [18F]FDG-PET Acquisition Times in Giant Cell Arteritis"

_diagnostics, 2023, doi:10.3390/diagnostics14010062_

Round 1

Reviewer 1 Report

Comments and Suggestions for Authors

1. Figure 2 Add arrow in maxillary arteries and temporal arteries. How about the vertebral artery?

2. Small modification in the file. 

    Line 115, 116.

    Line 436, 437

Author Response

Dear reviewer,

Thank you for your comments. We have changed the manuscript according to your comments. We agree that this improves the manuscript. Below, we respond to your individual comments and explain what we changed.

  1. Figure 2. Add arrow in maxillary arteries and temporal arteries. How about the vertebral arteries?

We would like the thank the reviewer for this suggestion and agree that adding arrows would make the figure more clear. Therefore, we added arrows for the maxillary and temporal arteries. Furthermore, we added a images showing the vertebral arteries and included arrows here as well.

  1. Small modification in the file:

Line 115, 116:  What was the aquisition time per table?

We would like to thank the reviewer for this relevant question. We use the continuous table motion method as described in the manuscript. All patients were injected with 2-3 MBq/kg [18F]FDG and 60 minutes post-injection underwent [18F]FDG PET/CT whole body imaging with LAT of the head with 0.3 mm/sec table motion and SAT of the head, neck and whole body region with 1 mm/sec continuous motion of the patient table. Compared to a step-and-shoot scan method, the acquisition time of the continuous table motion method with LAT was 13.8 minutes and the acquisition time of the SAT of the head, neck and whole body was 16.6 minutes per table. These times were calculated using the average head length of 25 cm: 250/0.3/60 = 13.8 minutes per LAT trajectory of the head and 1000/1/60 = 16.6 minutes for the SAT trajectory head to groin. We added this extra information in the method section of our manuscript (lines 116-120 of the revised manuscript)

Line 436, 437: We added references here.

Reviewer 2 Report

Comments and Suggestions for Authors

I read with interest your paper validating the LAT approach in diagnosing GCA.  It has relevance in the daily activity.

No remarks to do, even if more examples of LAT and SAT scan could have been more appreciated.

Author Response

Dear reviewer,

Thank you for your comments. We are glad to read that you see the relevance of this study. We also agree on the added benefit of an extra image. We added an extra figure showing the vertebral artery to show another example comparing LAT and SAT.